# Bacteria evoke alarm behaviour in zebrafish

Joanne Shu Ming Chia [1], Elena S. Wall[2], Caroline Lei Wee [3], Thomas A.J. Rowland [1,5], Ruey-Kuang Cheng[1], Kathleen Cheow [1], Karen Guillemin [2,4] & Suresh Jesuthasan[1,3]

When injured, fish release an alarm substance (Schreckstoff) that elicits fear in members of their shoal. Although Schreckstoff has been proposed to be produced by club cells in the skin, several observations indicate that these giant cells function primarily in immunity. Previous data indicate that the alarm substance can be isolated from mucus. Here we show that mucus, as well as bacteria, are transported from the external surface into club cells, by cytoplasmic transfer or invasion of cells, including neutrophils. The presence of bacteria inside club cells raises the possibility that the alarm substance may contain a bacterial component. Indeed, lysate from a zebrafish *Staphylococcus* isolate is sufficient to elicit alarm behaviour, acting in concert with a substance from fish. These results suggest that Schreckstoff, which allows one individual to unwittingly change the emotional state of the surrounding population, derives from two kingdoms and is associated with processes that protect the host from bacteria.

[1] Lee Kong Chian School of Medicine, Nanyang Technological University, Singapore, Singapore. [2] Institute of Molecular Biology, University of Oregon, Eugene, OR, USA. [3] Institute of Molecular and Cell Biology, Singapore, Singapore. [4] Humans and the Microbiome Program, Canadian Institute for Advanced Research, Toronto, ON M5G 1Z8, Canada. [5]Present address: St. Edmund Hall, University of Oxford, Oxford, UK. Correspondence and requests for materials should be addressed to S.J. (email: sureshj@ntu.edu.sg)

A large number of freshwater fish display a dramatic alarm response when exposed to conspecifics that have been injured[1–3]. This change in emotional state, which is characterized by changes in locomotion, such as darting and freezing, involves the olfactory system[4] and appears to be an adaptation to an indicator of potential danger. The alarm substance has been proposed to be produced by club cells[5,6], which are unusually large cells located in the epidermis. This hypothesis is based on the observations that injury to the skin is sufficient to generate a response and that many species that produce an alarm substance possess club cells[6]. In addition, seasonal loss of club cells is correlated with loss of the alarm substance[7]. However, fish that have club cells do not necessarily produce a substance that causes fear in conspecifics[8,9], indicating that club cells may have some additional function. The findings that the size of club cells increases in the presence of pathogens[10,11] or polluted water[12], while suppression of the immune system by cortisol causes a reduction in club cell numbers[13], have led to the proposal that club cells function in epithelial stress[12] or innate immunity[10,13]. It is unclear how an immune function for club cells relates to the alarm substance.

One possibility is that the alarm substance originates from a compartment of the skin that is directly linked to immunity, with club cells having a secondary role, such as storage. Several lines of evidence indicate that the alarm substance can be found in mucus. An alarm response can be elicited by material generated by vortexing fish[14], a procedure that releases mucus. Biochemical characterization indicates that the alarm substance is a mixture that can be depleted from skin extract by wheat germ agglutinin (WGA), which binds mucus[15–17], or an antibody to chondroitin sulfate[14], which is a component of mucus[18]. However, mucus evokes a strong response only after it has been heated[14], implying that some form of breakdown is required. Mucus, which contains a diverse population of bacteria[19], has multiple functions in fish skin, including defence against infectious agents[20–22]. Although mucus is produced by goblet cells, components of mucus such as glycosaminoglycans[23,24] have been found inside club cells.

Here, we ask whether mucus is transported to club cells; given that bacteria are lysed by heat[25], we also ask whether bacteria are also transported to club cells and test the hypothesis that bacteria provide a component of the alarm substance. We find that both bacteria and mucus are trafficked to club cells, with cell invasion playing a role. Our data establish that a substance from bacteria can function as an alarm substance, and provide insight into the evolution of the alarm response in fish.

## Results

**Mucus is transported into club cells**. Club cells, which are large ovoid cells embedded in the epithelium (Supplementary Fig. 1), have no opening to the external surface[6]. We hypothesized that mucus may be trafficked indirectly from the surface to club cells. To test this, we performed a pulse chase experiment. Fish were transiently incubated in Alexa594-WGA to label mucus[15,16], and imaged at various times subsequently to determine the fate of mucus. We used the nucleic acid dye SYTO 9 to label the epithelium. Except for the nucleus and perinuclear organelles, the cytoplasm of club cells is unlabeled by this dye and appears as dark areas; other cells, in contrast, are broadly labeled, with some heterogeneity in label according to cell type. After 2 h incubation in Alexa594-WGA, label was seen on surface epithelial cells, enriched on the apical ridges that have a high concentration of mucus, in puncta within superficial epithelial cells (as previously reported for larval periderm[15]), and goblet cells (Fig. 1a, b; Supplementary Movie 1; see Supplementary Fig. 1 for schematic diagram of zebrafish skin). No label was seen below the surface

layer ($n = 3$ fish). Following a period of 1–2 days in clean water, label was still seen within all surface cells and in all goblet cells (Fig. 1c). In addition, WGA puncta were detected in deeper cells (Fig. 1d–f), and in 214/507 club cells (Fig. 1c; Supplementary Fig. 2; Supplementary Movie 2; $n = 4$ fish). This suggests that mucus can be transported from the surface into club cells.

The presence of WGA puncta first within epithelial cells and goblet cells, and only subsequently inside club cells is consistent with transcytosis, which is a phenomenon that has been documented in the mucosal epithelium of other fish[26]. In addition to puncta, a few club cells contained an additional labeled cell (Fig. 1g–i). Club cells in several fish species, such as catfish and carp, have been reported to contain additional cells[12,27–29], apparently as a result of cell entry. To verify that there can indeed be a second cell within club cells of zebrafish, we imaged zebrafish that had been labeled with SYTO 9, which labels nuclei acids and phalloidin, which labels cortical f-actin. As seen in Fig. 1j–o and Supplementary Movie 3, an additional cell can be found within the cytoplasm of zebrafish club cells. We noticed that the cytoplasm of some club cells that contained a second cell was diffusely labeled with Alexa594-WGA (Fig. 1p–x; Supplementary Movie 4). This suggests that cell entry could contribute to the presence of mucus within the cytoplasm of club cells.

Invasion giving rise to the cell-in-cell phenomenon has been reported in a number of tissues, including epithelia[30,31]. Invasion is an active process and contrasts with phagocytosis, which involves uptake of an apoptotic cell. Consistent with active invasion, we could detect internalized cells that were unlabelled by an antibody to cleaved Caspase-3 (Fig. 2a, b). There are several different forms of invasion, such as entosis[32] and emperipolesis[33]. We first examined whether entry into club cells could involve entosis, a process that is dependent on cadherin-mediated adhesion, characterized by an accumulation of f-actin at the rear end of in the invading cell[34], and can lead to death of the intruding cell by autophagy in a noncell autonomous manner. No accumulation of β-catenin was seen at the interface between club cells and invading cells (Fig. 2c; $n = 4$ fish), and no accumulation of actin was seen at the rear end of the entering cell (Fig. 2d–i), while LC3B label was detected within the internalized cell, rather than surrounding it (Fig. 2j). Another feature of entosis is that it involves homotypic internalization. We noted that the nuclear morphology of the internalized cell differed from that of club cells (Fig. 1j, k), indicating that this is heterotypic entry. Thus, entry into club cells may involve mechanisms distinct from entosis.

Further imaging indicates that the internalized cells are motile (Fig. 2k–m; Supplementary Movie 5; Supplementary Fig. 3), which is a feature of other forms of invasion, such as emperipolesis. Internalized nuclei were negative for ΔNp63[35], which was detected in most epithelial cells (Fig. 2n–p). To test whether the cells include neutrophils, which are motile and invasive cells, we examined the *mpx:eGFP* line[36], in which neutrophils express eGFP under the control of the myeloperoxidase promoter. Neutrophils were strongly fluorescent in these fish (Fig. 2q), as were the cytoplasm of scattered club cells (Fig. 2r, s; Supplementary Movie 6; $n = 3$ fish). We thus propose that one cell type that can transfer material to club cells is neutrophils.

**Bacteria are transported to club cells**. Bacteria are known to produce substances that can stimulate chemosensory systems of vertebrates[37–39]. They are found within mucus of fish skin, and once inside the fish, can be engulfed by neutrophils[40]. These properties make bacteria potential components of an alarm signal that is transported from the surface into club cells. To test whether bacteria can be transported like mucus, fish were placed overnight in water containing chemically killed *E. coli* that had

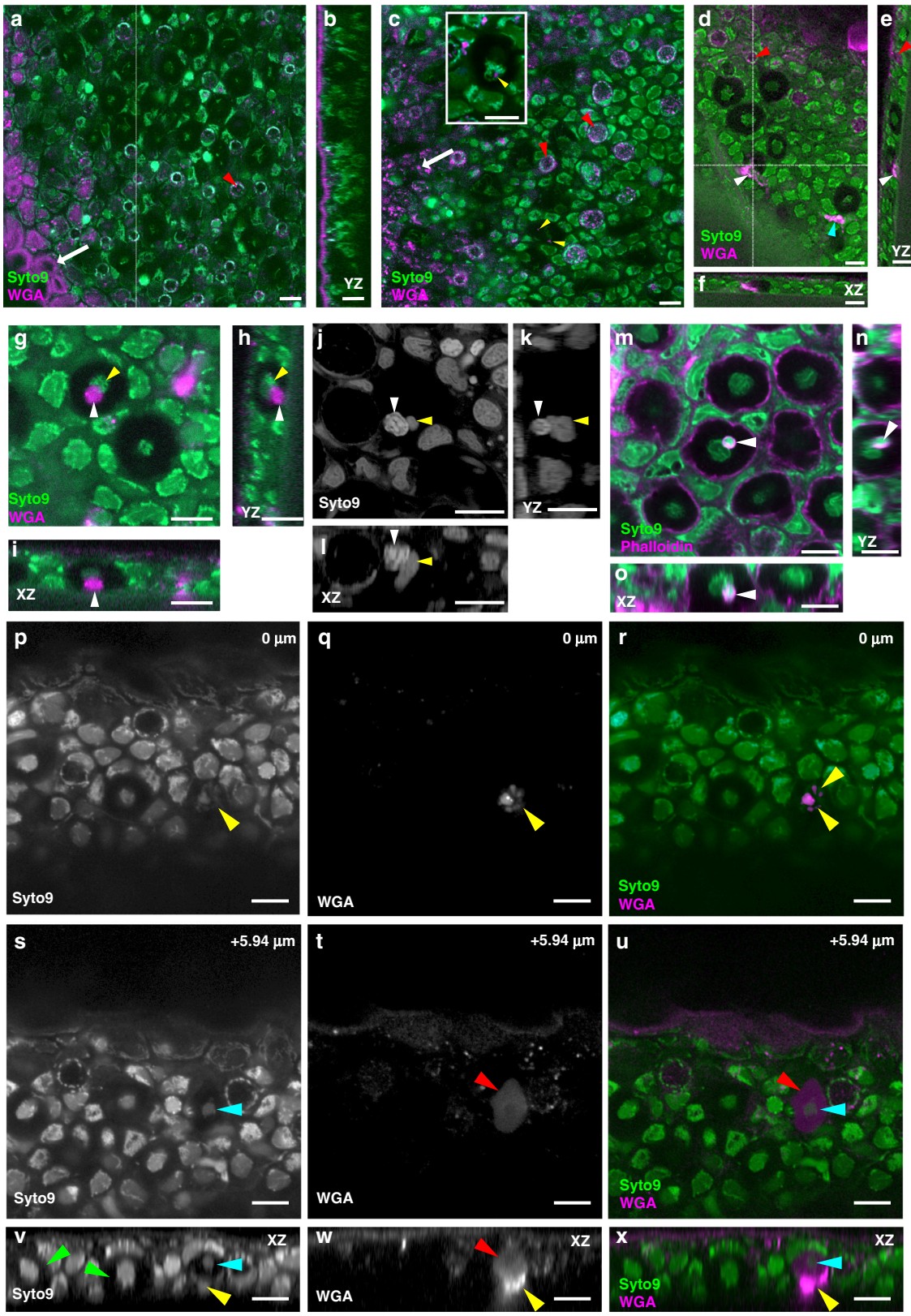

been labeled with a fluorescent dye pHrodo Red. The following day, punctate label was found in surface epithelial cells, goblet cells, neutrophils and club cells (Fig. 3a–e; n = 6 fish). Diffuse label was detected throughout the cytoplasm of a subset of club cells in all fish imaged (11.6% ± 6.6%; average ± standard deviation; n = 596 cells; Supplementary Fig. 4; Fig. 3a;

Supplementary Movie 7). Repeated imaging suggests that this diffuse label is the result of transfer of material from an invading cell (Fig. 3f, g). Uptake into goblet cells, neutrophils and club cells was also seen with killed *Staphylococcus aureus* (Fig. 3h, i; n = 3 fish; Supplementary Movie 8) and zymosan (Fig. 3j, k; n = 3 fish). These results demonstrate that bacteria and other

**Fig. 1** Mucus is transported from the surface into club cells. **a** Oblique section through the skin of an adult zebrafish, after 2 h of incubation in Alexa594 WGA (magenta). Cells are visualized by SYTO 9 (green), which binds nucleic acids. WGA binding is visible on the surface (arrow), and in goblet cells (red arrowhead). **b** Reconstructed transverse section through a z-stack of the fish in **a**. WGA (magenta) is restricted to the surface. **c** Oblique section through the skin of a fish, after 2 h label in WGA followed by 2 days in clean water. Punctate label is visible in superficial cells (white arrow), goblet cells (red arrowheads) and in club cells (yellow arrowheads; see inset). **d** Oblique section through another fish after transient label with WGA and a 2-day chase, showing label in elongated cells (white and cyan arrowheads). **e, f** Reconstructed transverse sections through the z-stack of the fish in **d**, showing the location of the labeled cell (white arrowhead) below a club cell. A superficial goblet cell is indicated (red arrowhead). **g** An Alexa594 WGA labeled cell (magenta; white arrowhead) inside a club cell. The club cell ER is indicated by the yellow arrowhead. **h, i** Reconstructed transverse sections through the club cell with the second cell. **j** A SYTO 9 labeled fish. The white arrowhead indicates a second cell inside the club cell. The club cell nucleus (yellow arrowhead) is morphologically distinct. **k, l** Transverse sections through the club cell. **m–o** Alexa-568 Phalloidin labeling to visualize f-actin. The club cell contains a second cell (white arrowhead) with cortical labeling. **n, o** Transverse sections through the club cell. **p–x** Two different focal planes of the skin of a zebrafish, following transient label of mucus by Alexa594-WGA and 2 days incubation in clean water. **p–r** show a deeper plane of focus, with a WGA-containing cell that may be blebbing (yellow arrowheads); **s–u** show a shallower plane with the club cell nucleus visible (cyan arrowhead). **t** The cytoplasm of the club cell is filled with Alexa594 WGA (red arrowhead). **v–x** Reconstructed transverse view through the z-stack. The club cell nucleus is indicated by the cyan arrowhead, while the WGA labeled cell is indicated by the yellow arrowhead. Club cells without internalized cells are indicated by the green arrowhead. Scale bar = 10 µm

particles can be taken into the skin of zebrafish and transported to club cells.

To test whether endogenous bacteria show a similar distribution, we performed in situ hybridization with an oligonucleotide probe to conserved sequences of bacterial 16S rRNA[41]. Label was seen in cells resembling neutrophils in all fields of view (Fig. 3l; $n$ = 27 images in 11 fish; 3.6 ± 2.5 labeled cells (average ± SD) per 156 µm × 156 µm area). Club cells containing a second cell with associated puncta (Fig. 3n–q; Supplementary Movie 9), or that was fully labeled (Fig. 3s), were also detected. The pattern is consistent with the hypothesis that endogenous bacteria are transported with mucus into club cells, potentially involving invasion of neutrophils.

**Bacterial lysate elicit a change in behavior.** Given the presence of bacteria in mucus and in club cells, we asked if bacteria could contribute to the alarm substance. To that end, we used a panel of diverse bacteria, including Gram-negative and Gram-positive strains that had been isolated from laboratory zebrafish and for which complete genome sequences have been determined[42], and one human isolate (see Supplementary Table 1). The bacteria were grown in culture, and then lysed by heating and the bacterial lysates were tested on individual fish. Of the ten strains tested, one strain—the Gram-positive *Staphylococcus saprophyticus* (ZWU0021)[42]—evoked a prolonged freezing response in two out of four tester fish (Fig. 4a, b, e; Supplementary Fig. 5). This bacterial lysate also caused fish to move to the base of the tank, which is typical of alarm behavior[14] (Fig. 4c, d, f; Supplementary Fig. 6). Other bacterial lysates (*Pseudomonas aeruginosa*, *Kocuria sp.* and *Aeromonas sp.*) gave a weaker effect, e.g., freezing for shorter durations or movement to the tank base without freezing, whereas lysates from the remaining bacterial strains did not elicit any obvious change in behavior. Darting behavior was limited (Fig. 4g). To obtain an indication of the responsiveness of the tester fish that did not react to the lysate (to rule out false negatives), nonresponders were exposed to skin extract. 20 out of 30 such fish froze to skin extract, demonstrating that they could detect the alarm substance. These observations establish that lysis of a subset of commensal bacteria releases a substance that is capable of eliciting behaviors that are characteristic of the alarm response.

Detection of the alarm substance has been shown to involve the olfactory system. We asked whether bacterial lysates can signal via the same olfactory receptors as the skin-derived substance. In general, each olfactory sensory neuron expresses a single or a few receptors, and all neurons with the same receptor combination project to the same glomerulus. To compare receptor binding of

fish extract and bacterial lysates, we thus imaged glomeruli. Previous imaging of larval zebrafish with broad expression of a calcium indicator suggested that the active fraction of skin extract strongly activates a dorsolateral glomerulus[14]. The responding neurons include microvillous neurons, as skin extract evoked a response in the dorsolateral bulb of *Tg(trpC2:GAL4, UAS:GCaMP6s)* (Fig. 5a, b) but not *Tg(OMP:GAL4, UAS:GCaMP6s)* fish. A response to skin extract was also seen in the *Tg(gng8:GAL4, UAS:GCaMP6s)* line, which labels a subset of microvillous OSNs[43] (5c, d). Lysates from commensal bacteria did not activate OSNs innervating the dorsolateral glomerulus (Fig. 5e, f), but did elicit a response in OSNs terminating more ventrally (Fig. 5g, h, i). A survey of the response of lysates from several different strains indicated that *Staphylococcus sp* ZWU0021 elicited a response that was more similar to skin extract, compared with another Gram-positive bacterium *Kocuria sp* ZOR0020, or to the Gram-negative *Aeromonas sp* ZOR0001 (Fig. 5j).

The absence of a response in the dorsolateral glomerulus to the bacterial lysate raises the question of whether bacteria are the sole source of the alarm substance. One potential method to address this could be to test the effect of skin extracts from adult fish that have been treated with antibiotics. However, this treatment is relatively inefficient at eliminating bacteria. Although there is a reduction in the bacteria population in the skin (Supplementary Fig. 7a–d), some bacteria remain, and the population re-establishes itself relatively quickly (Supplementary Fig. 7e, f). Also, any bacteria derived products that are stored in the fish would not be eliminated with this approach. Thus, a better method would be to use fish that are germ-free from embryonic stages. We opted to work with larval fish, as obtaining healthy germ-free larval fish is more straightforward, given that there is no need for an external food source[44]. We first showed that lysates prepared from fish aged 7–8 dpf were able to evoke alarm behavior in adult zebrafish (Supplementary Fig. 8a–c), and elicit activity in the dorsolateral glomerulus (Supplementary Fig. 8d, e), indicating that an alarm substance can be derived from larvae. This set the stage for comparing the effects of lysate from germ-free versus conventional fish.

Like lysates from conventional fish, lysates prepared from germ-free larval fish could elicit alarm behavior (Fig. 6a), as judged by increased darting (Fig. 6d) and time spent at the tank base (Fig. 6c). The mean increase in time spent darting with germ-free lysate was 7.33s [95%CI 2.33, 12.3], similar to the time of 8.83 s [95%CI 4.67, 12.2] evoked by conventional lysate. The proportion of time spent in the lower third of the tank after lysate delivery, compared with before lysate delivery, increased by 0.318 [95%CI 0.107, 0.467] for germ-free fish; this is similar to the

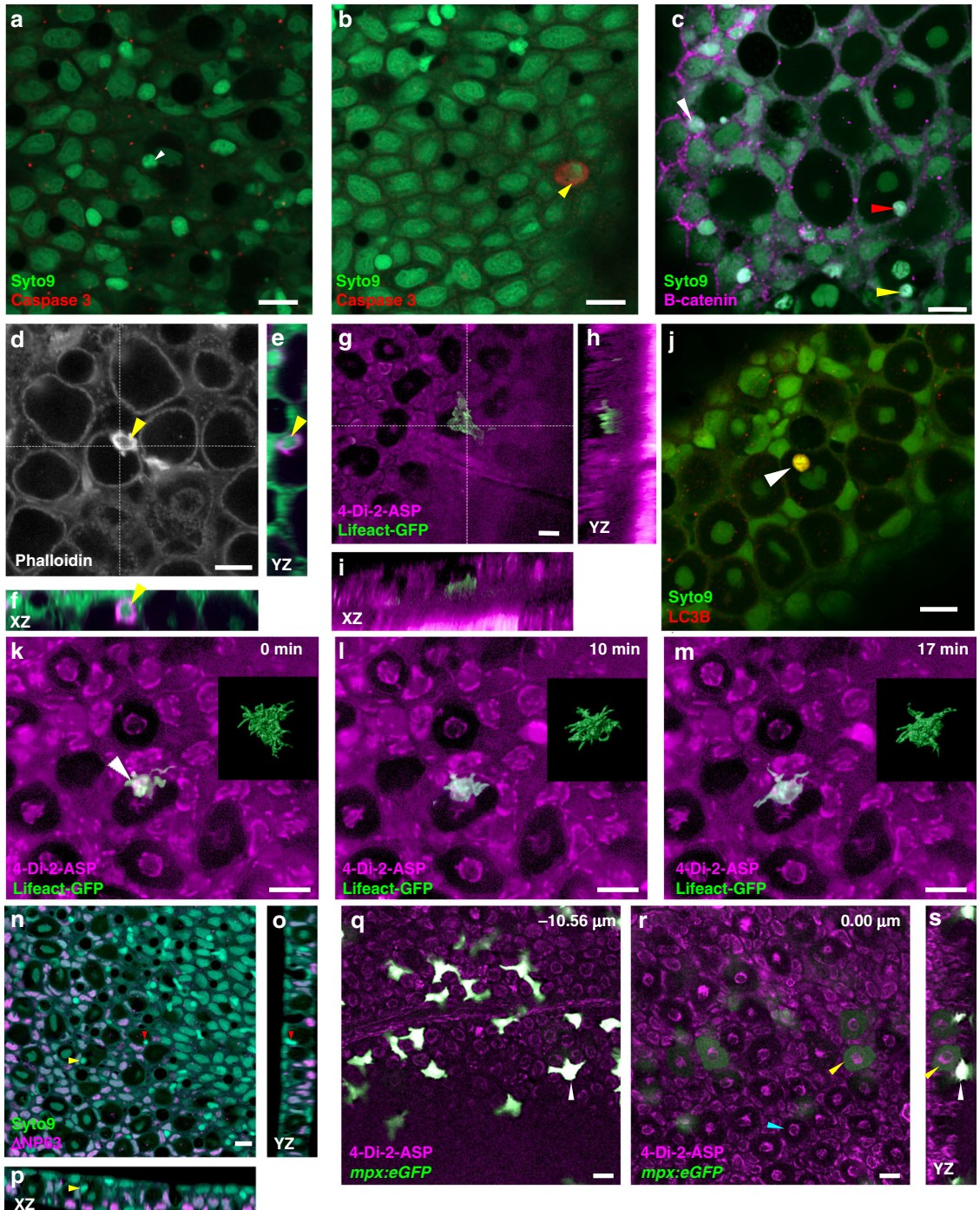

**Fig. 2** Characterization of cell invasion in zebrafish skin. **a**, **b** Labeling for cleaved Caspase-3. **a** The second cell within a club cell (white arrowhead) is unlabelled. **b** An example of a labeled skin cell (yellow arrowhead). **c** Label for β-catenin. In contrast to superficial epithelial cells, which have an enrichment of β-catenin at the periphery (white arrowhead), no enrichment is detectable in cells entering (red arrowhead) or within (yellow arrowhead) club cells. **d** Phalloidin label for f-actin. A strongly labeled cell (yellow arrowhead) is visible at the edge of a club cell. **e**, **f** Transverse sections through cell in **d**, showing the peripheral position of the cell relative to the club cell. **g** A fish expressing Lifeact-GFP in a motile cell that appears to be extending a protrusion into a club cell. **h**, **i** Transverse sections through the center of the club cell. The skin was labeled with 4-Di-2-ASP, which leaves most the club cell cytoplasm unlabelled. The invading cell is located near the base of the epithelium. **j** Label for LC3B is detected within the second cell (white arrowhead), but not surrounding it. **k**–**m** Three different time points of the skin of a fish expressing Lifeact-GFP in cell that appears to be mostly inside a club cell (arrowhead), but with protrusions extending outside. The insets show surface rendering of the green fluorescence across the z-stack. The cell changes shape over the period of imaging. **n**–**p** Label for ΔNp63 (magenta). No signal was detected in cells invading club cells (yellow and red arrowheads). **q**, **r** Two different focal planes of a fish with neutrophils expressing GFP under the control of the *mpx* promoter. Neutrophils are strongly fluorescent. The cytoplasm of a subset of club cells is weakly labeled; the cyan arrowhead indicates an unlabelled club cell. **s** Transverse section through a labeled club cell (yellow arrowhead) and neutrophil (white arrowhead). Scale bar = 10 μm

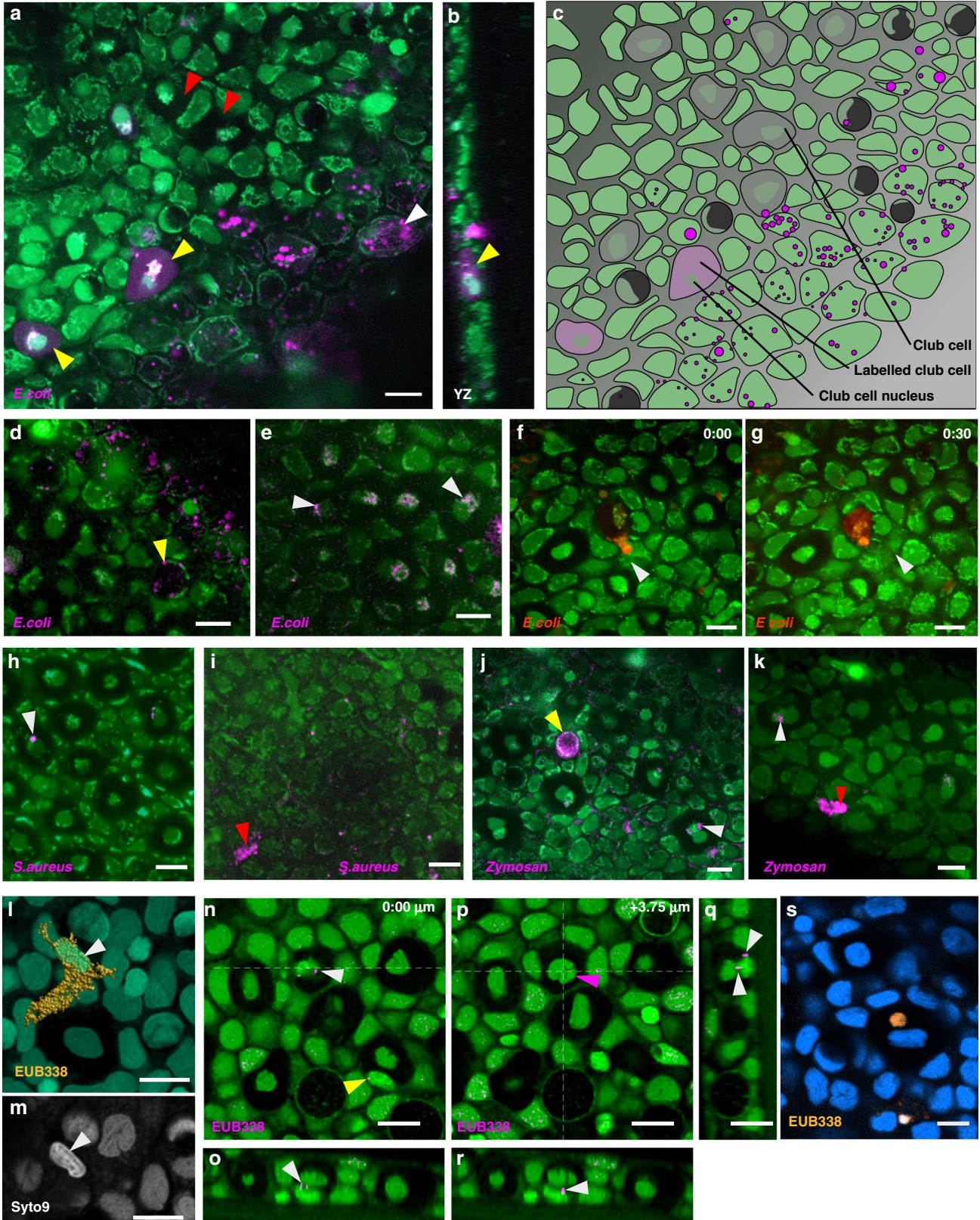

change of 0.263 [95%CI −0.181, 0.593] for conventional fish. Therefore, both germ-free and conventional lysates are able to evoke darting and bottom-dwelling behavior. In contrast, the extent of freezing was lower for germ-free lysate (Fig. 6b), with conventional lysate causing an average of 204.5 s freezing [95%CI 172.3, 228.5] compared with the 59.8 s [95%CI 8.0, 164.0] evoked by germ-free lysate. To further examine the effect of bacteria, the effect of lysates from germ-free fish was compared with that of conventional fish and germ-free fish that had been monoassociated with *Staphylococcus* ZWU0021. Again, germ-free lysate induced less freezing (Fig. 6f). These observations demonstrate that a fish component can evoke alarm behavior, particularly

**Fig. 3** Bacteria are transported to club cells. **a** An oblique section through the skin of a fish after overnight incubation in pHrodo Red labeled *E. coli*. The cytoplasm of two club cells is diffusely labeled (yellow arrowheads); other club cells do not have this diffuse pHrodo red fluorescence in their cytoplasm (red arrowheads). See Supplementary Movie 7. **b** A transverse section through the z-stack in **a**, showing one labeled club cell (yellow arrowhead). **c**. A schematic diagram of the slice in panel a, showing the different cell types. **d** Oblique section through the skin of another fish after overnight incubation in pHrodo Red labeled *E. coli*. Puncta are visible in goblet cells (arrowhead). **e** Localization of pHrodo Red labeled *E. coli* to perinuclear organelles of club cells (arrowheads). **f, g** Entry of a cell carrying pHrodo Red labeled *E. coli* into a club cell. **f** The cell (arrowhead) is initially adjacent to the club cell, but appears to have transferred some of its cytoplasm into the club cell. **g** After 30 min, the cell appears to be inside the cytoplasm of the club cell, which is now more intensely labeled. Localization of pHrodo red labeled *S. aureus* (**h, i**) and zymosan (**j, k**) within the epithelium, after transient incubation. **l** A neutrophil with bacteria, visualized by the EUB338 probe. Fish cells are labeled with SYTO 9 (green). This is a surface rendering of a z-stack, showing all focal planes. **m** A single plane of the stack in panel l, showing SYTO 9 label. The nucleus of the neutrophil (arrowhead) resembles the cell seen inside the club cell in Fig. 1e. **n, p** Two different focal planes of a region of the skin following in situ hybridization with the EUB338 probe. **n** At a deeper plane, several puncta are visible (white arrowhead), associated with a nucleus. One puncta is also associated with a second cell in a neighboring club cell (yellow arrowhead). **o** Transverse section through the club cell in **n**, with two puncta visible (arrowhead). **p** A shallower focal plane, showing the characteristic bi-lobed nucleus of a club cell (magenta arrowhead). **q, r** Transverse sections through the z-stack in **p**, showing labeled puncta associated with the deeper cell (arrowheads). **s** A rounded cell inside a club cell, labeled with probe to 16S rRNA. Scale bar = 10 μm

darting, independent of bacteria, but also suggest that the alarm substance is most effective when both bacteria and fish are present, consistent with previous reports that the alarm substance is a mixture[14,45].

## Discussion

Club cells in the skin of fish have been proposed to function in immunity[10] and also in production of the alarm substance[6,7]. The data here are consistent with club cells having a role in host defence, and suggest that this is linked to storage of the alarm substance. Superficial epithelial cells and goblet cells readily internalize mucus and bacteria that are on the surface, as indicated by experiments with labeled WGA, *E. coli* and *S. aureus*. These are then transferred to club cells. The process may involve transcytosis or transfer by immune cells, such as neutrophils; the possibility that transfer occurs via other cells cannot be discounted, as previous EM studies indicate that several different cell types can be found in club cells[28]. The reason for such transport is unclear at present, but an intriguing possibility is that club cells function at the interface between innate and adaptive immunity, possibly as antigen presenting cells. This could play a role in tolerance of commensal bacteria, as in the gut[46]. The ability of neutrophils to transfer their cytoplasmic contents has been reported in a few cases—for example, proteins can be transferred from neutrophils to endothelial cells[47] or to T cells[48]. In these cases, transfer requires direct contact, with the neutrophils remaining outside the target cell. In the zebrafish skin, transfer can begin when neutrophils are outside club cells, as indicated by observations in the fish containing labeled *E. coli*. However, cell invasion also occurs, and may represent an additional mechanism by which material from the surface is delivered to club cells. Given the motility of the internalized cell, some invasion may occur via a process resembling emperipolesis[49,50].

In principle, any substance that enters surface epithelial cells or goblet cells can be re-exported and transported to club cells, where it is accumulated and can serve as an alarm substance. It has previously been shown that an alarm response can be evoked by mucus that had been briefly digested with chondroitinase ABC[14]. Although this was interpreted as evidence for chondroitin fragments being a component of the alarm substance, chondroitin sulfate only elicited weak behavior effects. An alternative interpretation of the data is that other molecules, such as metabolites that are present in mucus[51], may be transported to club cells. When club cells are ruptured, these could act together with the bacterial component as Schreckstoff. It is notable that the bacterial lysates evoked freezing and movement to the tank base, but did not cause much darting. This is in contrast to the extract from germ-free fish, which caused darting, but not much freezing. In

previous work, we had shown that different components of the alarm substance contribute to different aspects of the behavior[14] raising the intriguing hypothesis that freezing may be evoked by bacteria, whereas the fish-specific component of Schreckstoff evokes darting. These components may act via different circuits. For example, the fish-specific component may act via olfactory neurons terminating in the dorsolateral bulb, as indicated by calcium imaging. While bacterial lysate can also be detected by the olfactory system, we speculate that they may additionally be sensed by other chemosensory neurons, such as solitary chemosensory cells, which are tuned to mucus in a species-specific manner and innervate the brainstem to provide an alternative neural pathway for defensive behavior[52].

There has been some debate as to how production of the alarm substance was selected for in evolution, as the substance was released only upon injury or death of the sender, who would derive no benefit[53]. Kin selection appears to be uninvolved, as responding fish are not necessarily related to the sender[54]. Better support was provided for the predator attraction hypothesis, which proposes that the substance attracts secondary predators and provides an opportunity for the sender to escape[55,56]. However, it was uncertain how effective this could be in practise[57], and not all predators respond to the alarm substance of prey[58]. The data here suggest that the alarm substance could have evolved as a result of processes involved in mucosal immunity. Accumulation of mucus and bacteria inside club cells would lead to a build-up of the material that is released only upon injury, i.e., a suitable proxy for danger. This view is consistent with the correlation between the amounts of alarm substance and number of club cells[7]. The observation that whole larvae that have no club cells are still able to produce an alarm substance[59] may be due to the presence of mucus and bacteria in the skin and the gut.

The ability of bacteria to influence the vertebrate nervous system is well established, especially via the gut–brain axis. There is some evidence that bacteria could contribute to social olfactory signaling in other species (e.g., hyenas[60]), but a causal relationship in vertebrates has not been demonstrated. The data here establish that substances from commensal bacteria from one individual are capable of acutely influencing the emotional state of others. Instead of originating only from a dying host, i.e., serving a purely altruistic role, we suggest that the alarm signal derives from mechanisms that protect the host. The production and response to Schreckstoff thus involve processes that are beneficial to the individual and to the species.

## Methods

**Ethical statement**. The experiments described here were carried out in accordance with the guidelines and protocols approved by the IACUC of the Biological

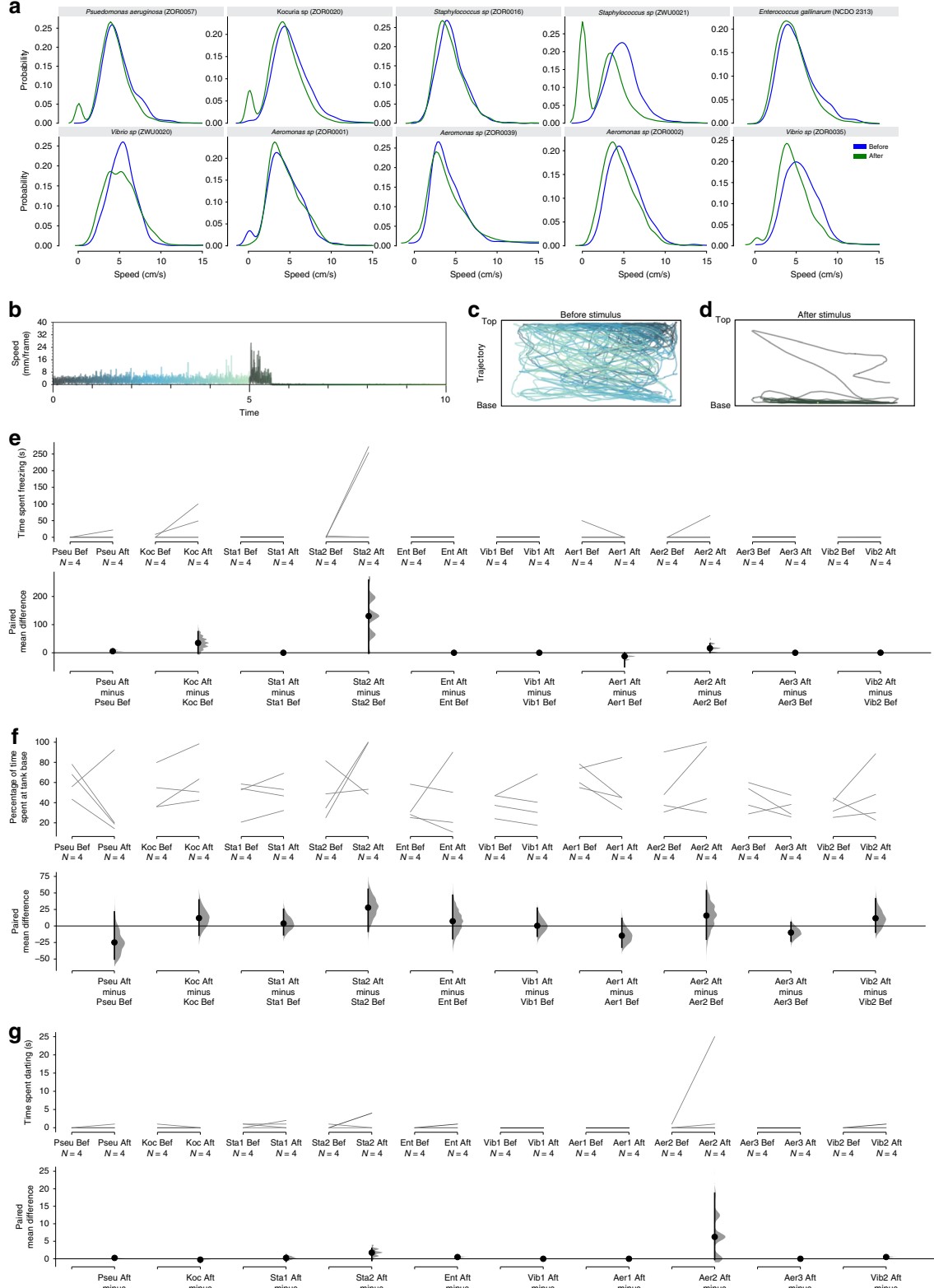

**Fig. 4** Effect of commensal bacteria on zebrafish behavior. **a** Effect of lysates from ten different cultured bacteria on behavior of adult zebrafish, as indicated by probablity density function of speed before (blue) and after (green) application of the lysate. **b**–**d** Response of one fish to lysate from the bacteria *Staphylococcus sp.* ZWU0021. **b** Speed of the fish. The trace is color coded according to time, with dark gray indicating the delivery of the lysate. Position of the fish, before (**c**) and after (**d**) stimulus. Effect of bacterial lysate of adult fish, as indicated by time spent freezing (**e**), percentage of time at tank base (**f**), and time spent darting (**g**), as shown by Cumming estimation plots[72]. The raw data are plotted on the upper axes and each paired set of observations (before (Bef) and after (Aft) delivery of bacterial lysate) is connected by a line. On the lower axes, each paired mean difference is plotted as a bootstrap sampling distribution. Mean differences are depicted as dots; 95% confidence intervals are indicated by the ends of the vertical error bars

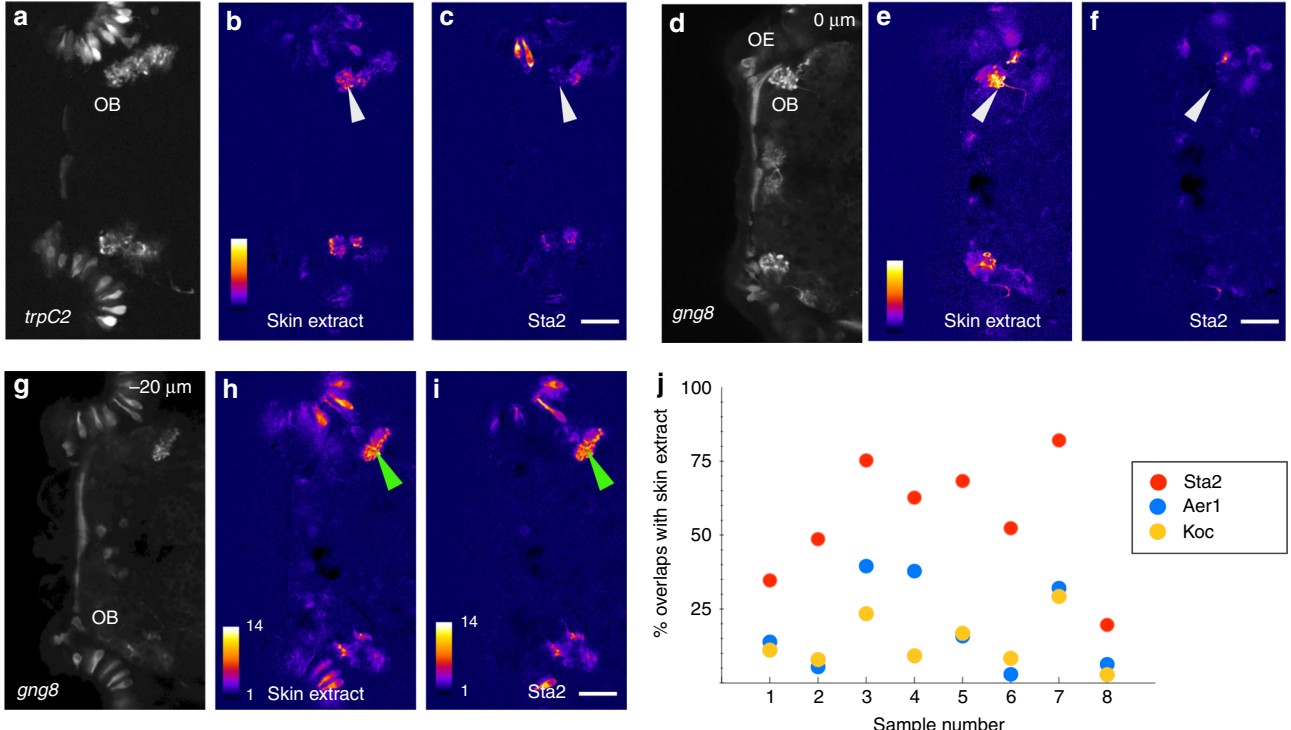

**Fig. 5** Effect of bacterial lysate on the olfactory system. **a–i** Calcium imaging of the olfactory bulb two different larval fish exposed to skin extract (**b**, **e**, **h**) or bacterial lysate (**c**, **f**, **i**). Average projection of the time-series, showing GCaMP6s expression in olfactory sensory neurons and in terminals, driven by the *trpc2* (**a**) or *gng8* (**d**, **g**) promoter. Panels **d–i** are the same fish, with **g–i** showing a plane that is 20 μm deeper than **d–f**. A glomerulus in the dorsal bulb (white arrowhead) is activated by skin extract but not by the bacterial lysate. Both lysates activate identical terminals in the ventral bulb (green arrowhead). **j** Percentage of overlap between response to three different bacterial lysates and skin extract. The highest overlap is seen with lysate from *Staphylococcus sp.* ZWU0021. Anterior is to the left; OB olfactory bulb, OE olfactory epithelium. Scale bar = 25 μm

Resource Center at Biopolis, Singapore (#151092) and by the IACUC of the University of Oregon (#15–98).

**Fish lines**. The transgenic lines used were: *TgBAC(gng8:GAL4FF)c426*[61], *Tg(trpC2:GAL4)p209*[62], *Tg(OMP:GAL4FF)rw0320*[63], *Tg(UAS:GCaMP6s)sq205*, and *Tg(actb2:LIFEACT-GFP)e114*. The *e114* line labels motile cells in the skin of adult fish (Supplementary Movie 10).

*Tg(nbt:jGCaMP7f)sq214* was generated using Gateway cloning. The coding sequence for jGCaMP7f was amplified from pGP-CMV-jGCaMP7f (a gift from Douglas Kim; Addgene plasmid #104483) using Phusion® High-Fidelity DNA Polymerase. This was placed downstream of the 3.8 kb *Xenopus laevis* neural beta tubulin (Xla.Tubb) promoter, in a Tol2 plasmid, and injected into embryos at the 1-cell stage together with Tol2 transposase mRNA.

**Culture and lysis of commensal bacteria**. Bacterial strains were obtained as described by Stephens et al.[42], except for *Enterococcus gallinarum*, which was obtained from ATCC (strain 49573). All strains were maintained in 25% glycerol at −80 °C. Most bacteria were directly inoculated into 5 mL Tryptic Soy Broth media (BD 211825) and grown for ~16 h (overnight) with shaking at 30 °C. *Kocuria sp*, *Staphylococcus* strains, and *Enterococcus gallinarum* were grown for ~16 h (overnight) in Brain Heart Infusion Broth media (BD B237500) at 30 °C with shaking, except *Staphylococcus sp* ZOR0016, which was grown stagnant.

Bacterial cultures were washed twice; once by centrifugation, aspiration of supernatant, and suspension in 0.7% NaCl, and a second time by centrifugation, aspiration of supernatant, and suspension in a volume of sterile water equal to starting volume. Cell solutions were heated at 90 °C for 30 min to lyse bacteria. Cell debris was pelleted by centrifugation and the supernatant was filter sterilized with a 0.2 μm filter. Lysates were stored at 4 °C.

*Staphylococcus sp* ZWU0021 (IMG genome ID 2526164573) was classified by BLAST searches with the *gyrB*[64] and *rpoB*[65] genes.

**Production of skin extract**. Fish were euthanized in iced water before being placed in a clean petri dish, where shallow cuts were made to both sides of the fish with a sharp razor (Sharpoint, 22.5° stab). The cut fish was put into a 2 mL tube with filtered E3 (5 mM NaCl, 0.17 mM KCl, 0.33 mM CaCl$_2$, and 0.33 mM MgSO$_4$) and placed on a rocking platform shaker for 5 min. After discarding the fish, the liquid was topped up to 2 mL with E3 and heated at 95 °C for 2 h. Heated extract was

placed on ice to cool before spinning at 10,000 rpm for 10 min. The supernatant was pooled from multiple fish before using in behavior assays.

**Generation of larval extract**. Lysates comprising unfed 7 or 8 dpf larvae were generated at a final concentration of 1 larva per 20 μL E3 media, on a laboratory benchtop under sterile conditions. First, a cell culture strainer immersed in a petri dish of sterile E3 media was used to concentrate the larvae. After a brief wash (i.e., transfer into another petri dish of sterile E3), larvae were moved to a 1.5 mL Eppendorf tube using a sterile transfer pipette, and anaesthetized on ice. After larvae had sunk to the bottom, most of the E3 was removed (leaving ~200 μL in the tube), and a sterile plastic pestle attached to a motor-driven grinder was used to thoroughly homogenize larvae for ~1 min The homogenized larvae were subsequently heated for 30 min on a 95 °C heat block (this heating step serves as a control for bacteria lysate experiments, but is unnecessary for the generation of an alarm response). Afterwards, the heated extract was centrifuged at 13,000 rpm for 5 min The supernatant was then transferred into a clean tube, and additional sterile E3 was added to make up the final volume of 1 larva per 20 μL E3. Larval lysates were stored at 4 °C until ready to use. They are effective for at least a few weeks.

**Behavior assay and analysis**. Testing of alarm behavior was conducted on TLEK wild types in a glass tank measuring 20 cm (L) by 5 cm (W) by 12 cm (H) containing 1 L of aquarium water. Naive fish more than 2 months old were habituated in the tank for 5–7 min before each test, and then recorded for 5 min before substance delivery and 5 min after substance delivery. Substances were manually pipetted into the tank, where 5 μL of test substance was diluted in 1 mL fish water before delivery. The alarm response is known to be highly variable and some batches of zebrafish do not respond reliably. Hence, as a control, skin extract was delivered to fish that did not respond to the test substance after the 10-min recording session. Typically, ~70% fish would respond to skin extract, and the entire set was discarded if the percentage of responders in a population was lower than 50%. No response was seen when fish water was delivered.

For video recording, tanks were placed in front of a black background with overhead lighting in a darkened room. Videos were recorded from the front at 30 frames per second by a Basler USB3 camera in a resolution of 1680 × 480 pixels for filming two tanks simultaneously. The position (*x–y* coordinates) of the fish was determined online based on background subtraction algorithms in Python utilizing OpenCV library. These data were then analyzed offline for speed and other measures. Fish speed was binned by cm/s over the 5-min period before and after

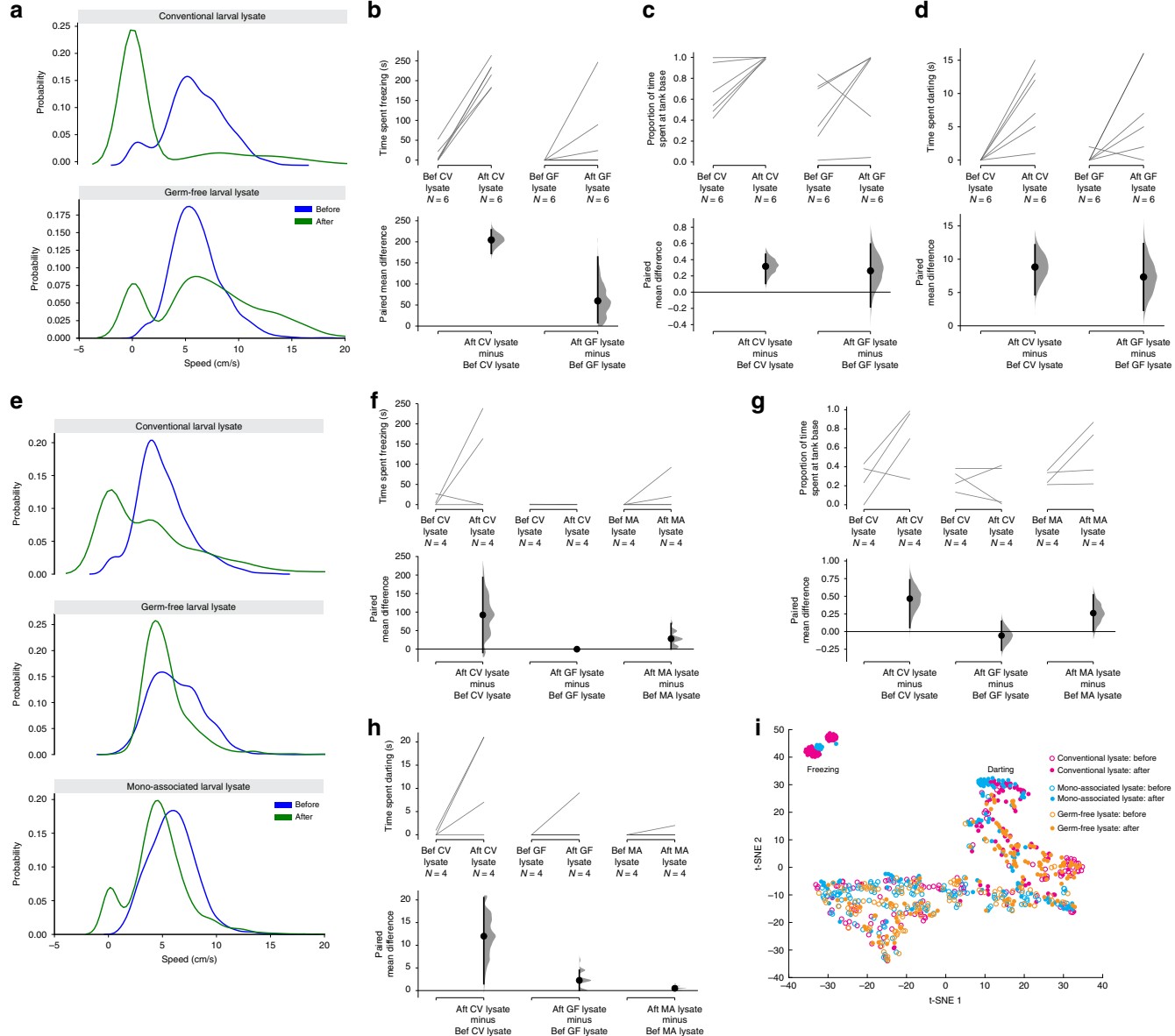

**Fig. 6** Fish and bacteria provide components of the alarm substance. Effect of lysate from conventional (CV) and germ-free (GF) larvae on behavior of adult zebrafish (n = 6 each), as shown by speed probability density function (**a**), and change in freezing (**b**), time spent at the tank base (**c**), and darting (**d**). **e–h**. Effect of lysate from a separate batch of conventional (CV), germ-free (GF), and monoassociated (MA) larvae on behavior of adult zebrafish (n = 4 each). **i** Swimming behavior was clustered using t-distributed stochastic neighbor embedding (t-SNE). Each data point represents behavior within a 5 s epoch. Filled circles indicate behavior that occurs after delivery of the lysate. The response to lysate from conventional is monoassociated larvae (blue and red circles) are more similar than the response to lysate from germ-free larvae (orange circles). The cluster on the top left corner corresponds to prolonged freezing behavior, which was not evoked by lysate from germ-free fish. Other postlysate clusters may represent darting behavior

the substance was delivered and plotted in a histogram. The probability density function for the before and after periods were superimposed to show the effect of the substance on fish speed.

For t-SNE-based clustering of behavior (MATLAB), eight features were extracted from $x$–$y$ coordinates: (1) Distance, (2) velocity, (3) acceleration, (4) absolute tank depth, (5) change in tank depth, (6) horizontal translation, (7) posture (i.e., width/height of bounding box), and (8) change in posture. The mean, minimum, maximum. and standard deviations of these features were then quantified across 5 s (150 frame) epochs for the entire length of the experiment (both before and after lysate presentation) and aggregated across 12 fish (four fish from each treatment of conventional, germ-free, and monoassociated fish lysates, respectively) in an $8 \times 4 = 32$ dimensional array. t-SNE was then applied for dimensionality reduction to two dimensions (MATLAB default parameters: https://www.mathworks.com/help/stats/tsne.html).

**Calcium imaging**. Two photons imaging at 1 Hz were carried out using a galvano scanner on a Nikon A1R-MP system attached to a FN1 microscope with a

25× water dipping objective and the Ti Sapphire laser tuned to 930 nm. Widefield imaging was performed using a Zeiss Axio Examiner with a 20× water dipping objective. Images were captured with a Hamamatsu Flash C4, controlled by MetaMorph. Illumination was provided by a blue LED (Cairn). Larval fish were anesthetized in mivacurium, and then embedded dorsal up in 2% agarose in E3 on a glass coverslip held in a Warner chamber. A wedge with an angle of ~120˚ was cut in front of the fish, which was then placed within in a diamond flow chamber (Warner R26). Stimuli were delivered using a perfusion device (Warner VC-6M) controlled by TTL pulses from the Elements or MetaMorph software.

Images were analyzed using FIJI software[66]. After registration to correct for $x$–$y$ drift, using the StackReg plug-in with the translation option, a running average of three frames was carried out. The change in fluorescence intensity (F/F0) was then obtained, using the ten frames prior to stimulus delivery as a baseline. Images show the frame with the maximum increase in fluorescence. For display, the "fire" LUT was used, with level set to a center of 5.69 and width of 10.67 in all images.

To compare the extent of OSN activation, the area of pixels with more than twofold increase in fluorescence was compared. The area of overlap between the

different stimuli was then measured, and represented as a percentage of the area with a response to skin extract.

**Adult antibiotic treatment**. Adult zebrafish from the same home tank were treated with a final concentration of 25 μg/mL rifampicin[67] for 5 or 18 h. The 50 mg/mL rifampicin stock was dissolved in DMSO before 1:2000 dilution in fish water. A 1 L crossing tank filled with 500 mL of rifampicin or fish water was used to treat fish. Fish were then netted with an autoclaved net and transferred into a 50 mL falcon tube containing filtered Tricaine (300 mg/L, for euthanasia). To obtain mucus layer, euthanised fish were transferred into a 50 mL tube containing 2 mL filtered E3 per fish and vortexed for 1 min with a break every 10 s. The supernatant was collected to test for the presence of bacteria by plating on LB or ARISA.

**Swab sample and DNA purification**. Fish were anesthetized in sterile-filtered E3 with Tricaine (Sigma, USA) and swabbed on both sides using sterile cotton swabs (LP Italiana, Italy). Each swab was incubated individually in 1.5 mL microcentrifuge tubes containing 300 μL of filtered E3, for 30 min at room temperature. Solutions were filtered through a Minisart® 0.2 μm syringe filter (Sartorius, Germany). Following incubation, the cotton swabs were squeezed of any remaining liquid and disposed. Each of the 1.5 mL microcentrifuge tubes was topped up to 1 mL with filtered E3 and then combined into a single tube. DNA was extracted and purified from 100 μL of the swab sample by treatment with sodium hydroxide, EDTA, and ethanol[68], which allows for extraction from both gram-positive and gram-negative bacteria.

**ARISA conditions and primers**. The universal primers used for ARISA PCR were 1406F: 5′-TGYACACACCGCCCGT-3′ and 23Sr: 5′-GGGTTBCCCCATTCRG-3′[69]. 2 μL of the purified DNA sample was used for each 25 μL reaction containing 1× Taq Buffer (Thermo Fisher Scientific, USA), 1 mM of MgCl₂, 2.5 U of Taq DNA polymerase, 0.2 mM (each) deoxynucleoside triphosphate (Promega, USA), and 0.25 μM (each) primer. The PCR conditions performed were as follows: 95 °C for 3 min, 30 cycles of 95 °C for 45 s, 55 °C for 1 min, and 72 °C for 2 min; and 7 min of final elongation at 72 °C.

**Generation of gnotobiotic larvae**. Methods described previously[44] were used to generate germ-free and monoassociated larvae. In performing behavioral experiments to assess synergism, we noted that the response to skin extract is dependent on the concentration of substance as well as the sensitivity of the fish. We diluted thus the lysates and compared fish against siblings grown in the same conditions.

**Uptake of fluorescently labeled bacteria and zymosan**. Fish measuring ~2 cm (~6 weeks old) were placed overnight in a beaker containing pHrodo Red labeled *E. coli*, *S. aureus*, or zymosan (Thermo Fisher P35361, A10010, P35364; 2 mg/50 mL E3). They were then transferred into clean water, labeled with SYTO 9 (1:5000), euthanized using MS222, and imaged using confocal microscopy. For long term imaging, fish were anesthesized and intubated as described[70].

**Mucus labeling**. Fish were incubated in Alexa Fluor 594 WGA (1:1000 dilution of a 1 mg/mL stock; Thermo Fisher W11262) for 2 h. To follow mucus uptake, fish were transferred into a tank with clean water for at least 15 h.

**In situ hybridization**. In situ hybridization was conducted with the EUB probe[71] (5′-GCT GCC TCC CGT AGG AGT-3′) labeled with Alexa Fluor® 546 (N-hydroxysuccinimide ester) at the 5′ end (IDT, USA). Four independent hybridizations were carried out. A second probe (5′-CGA CGG AGG GCA TCC TCA-3′) gave similar results. To control for autofluorescence in imaging, samples were processed in parallel without any probe. Two-month-old fish that were less than 2 cm in length were fixed in 4% paraformaldehyde in phosphate buffered saline (PBS). Samples were washed once in methanol (MeOH) for 10 min and then stored at −20 °C in MeOH for at least 2 h. Prior to hybridization, fish were rehydrated to PBS with decreasing concentrations of MeOH (75, 50, and 25%) in PBST (PBS with 0.1% Tween 20) for 5 min each. Samples were permeabilized in 10 μg/mL proteinase K for 5 min, and subsequently fixed for 20 min in PFA. After washing in PBST, hybridization was carried out with 5 μg/mL of the probe in hybridization buffer (2× saline sodium citrate buffer, 50% formamide) and incubated at 37 °C overnight. The samples were then washed in hybridization buffer once, followed by two washes 0.1 × SSC and then two washes in PBS.

Following in situ hybridization, nuclei were labeled by incubation in 1:5000 dilution of SYTO 9 (Thermo Fisher) for 1 h at room temperature. Fish were then held securely on the cover of a petri dish, using 3% low melting temperature agarose in PBS, and imaged on a Zeiss LSM800 confocal microscope with a ×40 1.0 N.A. water dipping objective. A pinhole of 1 Airy unit was used. Images form were median filtered (0.5) in Fiji, and contrast adjusted in Affinity Designer. Surface rendering was performed using Huygens Professional X11.

**Quantitation of bacteria label**. Z-stacks were maximally projected and the number of labeled cells was manually counted in FIJI.

**Antibody label**. Fish measuring ~2 cm in length were euthanized by overdose in tricaine (Sigma A5040), and then fixed at room temperature in PBS containing 4% paraformaldehyde. Following washing and permeabilization with PBS containing 0.1% Triton X-100, LC3B antibody (Abcam ab48934) was diluted 1:200 in PBS containing 0.1% Triton X-100 and 1% bovine serum albumin (Sigma A1470). For labeling β-catenin, the primary antibody (BD Bioscience 610153) was diluted 1:200, while cleaved Caspase-3 (Cell Signalling Technology 9661) and ΔNp63 (Santa Cruz SC-8341) were diluted 1:100. The secondary antibody, Alexa-568 goat anti-rabbit (Thermo Fisher A11011), or Alexa-647 goat anti-rabbit (Thermo Fisher A27040) was used at 1:1000. To label nuclei, fish were incubated in 1:5000 SYTO 9 (Thermo Fisher S35854) for 1 h. Three independent labeling experiments were carried out for each antibody.

**Actin label**. Following fixation and rinsing, fish were permeabilized with 2% Triton X-100 in PBS. Filamentous actin was labeled by overnight incubation in Alexa Fluor 568 phalloidin (Thermo Fisher A12380) at 1:20 dilution of a stock solution (0.2 units/μL) prepared in methanol.

**Reporting Summary**. Further information on research design is available in the Nature Research Reporting Summary linked to this article.

## Data availability
Data on animal behavior and confocal imaging are available on Figshare [https://doi.org/10.6084/m9.figshare.8796695].

## Code Availability
Software used to track fish and analyze their behavior is available on GitHub [https://github.com/rkcheng/AlarmSubstance_AdultFish].

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

## Acknowledgements

We thank Ajay Mathuru for help in optimizing the set-up for monitoring behavior and Sven Pettersson for discussion. This research is supported by the Lee Kong Chian School of Medicine, Nanyang Technological University Singapore Start-Up Grant (to S.J.), the Ministry of Education Singapore under award MOE2017-T2-058 (to S.J.), and by the National Institute of General Medical Sciences of the National Institutes of Health under award numbers P50GM098911 (to K.G.) and P01GM125576 (to K.G.). The content is solely the responsibility of the authors and does not necessarily represent the official views of the National Institutes of Health.

## Author contributions

J.S.M.C. performed behavior assays and analysis, ARISA and in situ hybridization. E.S.W. prepared lysates from germ-free and monoassociated larval zebrafish. C.W. devised the method for preparing effective lysates from larval zebrafish, prepared conventional and germ-free lysates, and analyzed behavior data. T.A.J.R. characterized cell internalization. R.K.C. wrote the software for the behavior assay and analyzed behavior data. K.C. performed qPCR to quantitate bacterial load and generated the *Tg(nbt:jGCaMP7f)sq214* line. K.G. designed experiments on using germ-free fish. S.J. conceived the project and designed experiments, performed calcium imaging, pulse chase of epithelial cells, tracing of bacteria, and wrote the manuscript. All authors edited and approved the manuscript.

## Additional information

**Competing interests:** The authors declare no competing interests.

