## [Peer Review File · Nature Communications]

Reviewers' comments:

Reviewer #1 (Remarks to the Author):

This work is designed to examine the role of epidermal club cell cannibalism and bacteria in generating fish alarm cues. I have worked on the Schreckstoff alarm system for the majority of my career and I am very excited by this current work. I am in favour of publication. I am primarily a behavioural and evolutionary ecologist, hence my comments are restricted to that component of the work. I have considered all of the techniques used by the authors, but I am not an expert in the methods used, hence I will defer to others to address whether the techniques are appropriate.

There are a couple of things that need to be changed prior to publication. This paper makes some rather bold statements in the abstract and the introduction (paragraph 2) that says there is no known benefit to the sender of Schreckstoff. This certainly was the case for many decades. Von Frisch identified fear substances in the late 1930's and the notion of no benefit to the sender did not change until we published 2 papers in *American Naturalist* (Mathis et al. 1995 and Chivers et al. 1996). We showed that predators were attracted to Schreckstoff (Mathis et al. 1995) and that when the second predators arrived, they would often fight with the original predator, leading to escape of the prey (Chivers et al. 1996). This secondary predator interference was clear experimental evidence of a benefit to the Schreckstoff sender. Others have documented similar findings of attraction to alarm cues (e.g. Wisenden, Lonnstedt). It is incorrect to say there are no benefits to the sender. My caution that secondary predator interference may be rare in the wild lead me to consider that alarm cues evolved in another context (i.e. that they were part of the innate immune system). Much of this was inspired by Iger et al. (1994; *Cell and Tissue Research*) who found photographic evidence of lymphocytes and leucocytes within club cells. The presence of two membranes around the lymphocytes and leucocytes indicates they were engulfed by the club cells, suggesting the involvement of club cells in the immune system.

You cited my original work on this (Chivers et al. *Proc B*) documenting that club cells are part of the immune system, but our other work (Halbgewachs et al. 2009, *Biological Journal of the Linnean Society*) was the most significant work linking alarm cells to immunity. Colin gave fathead minnows intraperitoneal injections of cortisol, a known immunosuppressant, or injections of a control substance (corn oil). He documented that fish exposed to cortisol had suppressed immune systems (as measured by a respiratory burst assay) and that they also reduced their investment in club cells. Until this current paper is published, this was the best evidence indicating that the club cells of Ostariophysan fishes are part of the innate immune system and that the alarm function of the cells evolved secondarily.

After altering the text appropriately, I would recommend publication of this work. Congrats on a great study.

Douglas P. Chivers
Distinguished Professor
University of Saskatchewan

Reviewer #2 (Remarks to the Author):

In this manuscript, Dr Jesuthasan and colleagues report that bacteria contribute to the zebrafish alarm response, acting synergistically with a substance from the fish. They suggest that superficial cells can internalise mucus and bacteria, and then invade club cells, in which the alarm substance (Schreckstoff) is made and stored. The authors propose that the formation of this cell-in-cell structure occurs through entosis and is required for the alarm response, concluding that 'skin cell cannibalism and processing of bacteria generate the zebrafish fear pheromone'.

The paper is clearly written and presented and the overall concept is of interest. I have been asked to provide a perspective on the data relating to cell engulfment, so will restrict my comments to lines 86-139, 255-270 and Figure 2. Although the notion of cell cannibalism could be interesting in this context, I have several major concerns, as outlined below.

1) Does cell-in-cell formation occur?

The authors conclude that club cells undergo cell-in-cell formation based on images in 1c, 1e and Fig 2.

While many of these images could be consistent with cell internalisation, they are certainly not definitive. To convincingly demonstrate cell-in-cell formation, it is necessary to rigorously confirm that the internalised cell is inside, rather than on or adjacent to, its host.

It would significantly strengthen these data to show:

a) Cell-in-cell formation in 3-D. A full confocal stack of a club cell hosting another cell would help confirm full internalisation. This would be best visualised using a marker that clearly defines each cell periphery in the pair (e.g. adherens junctions or plasma membrane).

b) An image acquired by electron microscopy to reveal how the internalised cell is housed inside the club cell – is there an obvious vacuole as would be expected during entosis?

In conclusion, more robust imaging data are required to prove cell-in-cell formation here. This would seem achievable, particularly as the penetration of club cells by other cell types has been reported previously by electron microscopy (Refs 15-18).

c) Can you explain why have more straightforward uptake mechanisms, such as transcytosis, been excluded as important in this context?

2) What is the mechanism of possible cell-in-cell formation?

Cell engulfment can proceed through a variety of mechanisms, including phagocytosis, emperipolesis and entosis. In their title, the authors use the term 'cell cannibalism', while in the text they infer that apparent cell-in-cell formation occurs through entosis (lines 103-114 and 270).

Entosis is a specific form of cell engulfment, with some key features including: 1) It is a form of live cell engulfment in which one viable cell is internalised into another, 2) It is a homotypic form of cell engulfment, occurring exclusively between paired epithelial cells, 3) Mechanistically, it depends upon adherens junction formation and localised actomyosin contractility (Overholtzer 2007).

Based on the data presented, it is unclear why the authors attribute apparent cell-in-cell formation to entosis, versus any other kind of engulfment? The data do not justify this conclusion, and there is no reason why this form of engulfment would be more likely or interesting than another.

a) Please can you clarify whether the club cell itself bears the typical features of an epithelial cell required for entosis (e.g. adherens junctions)?

b) Can you clarify whether all internalised cells are epithelial, as would be required for entosis? The majority of apparently internalised cells look small relative to the club cell – would this be expected of a superficial cell? Have you stained for an epithelial marker to be sure? Given that published work reports penetration of club cells by leucocytes, it would seem possible that heterotypic cell-in-cell formation may be occurring as well or instead (e.g. as can be seen between

epithelial and immune human cells).

c) Can you clarify whether the internalised cells are all live, as would be the case in entosis? Is it not possible that some of the examples shown could be phagocytosis of apoptotic cells?

d) The stills shown in Fig 2a-b appear to suggest quite rapid release of bacteria into a club cell as it contacts a labelled neighbour (a copy of the movie would be more helpful). This would be unexpected with respect to entosis, in which the internalised cell is scissioned into a sealed vacuole, which only later undergoes lysosomal degradation.

e) The statement in lines 105-6 that 'entosis is characterised by a higher level of f-actin in the invading cell relative to the host' is an oversimplification. What is important is spatiotemporally restricted activation of actomyosin contractility in the internalising cell, which pushes into its host. As such, a distinctive enrichment of actin, or phosphorylated myosin, can be seen specifically in the tail of the internalising cell (Purvanov 2014, Sun 2014). The overall intensity of actin staining is not very meaningful and certainly not diagnostic of entosis.

f) The data presented on LC3B in Figures 2k-l, and discussed in lines 110-114, have been misinterpreted. Autophagy is a recycling process that occurs in response to various stresses and is characterised by the formation of autophagosomes, often visualised as LC3-positive puncta. During entosis, a related but quite distinct, signalling pathway degrades engulfed material. In this case, LC3 is recruited directly onto the entotic vacuole. This process involves transient labelling at the surface of this large endocytic compartment with LC3 (Florey 2011), and looks nothing like the image presented in Figure 2k. Moreover, during entosis, LC3 recruits in the host cell, not within the inner cell, as shown in 2l.

g) Neither the actin nor LC3 data define this process as entosis. If the authors wish to explore entosis they should instead investigate i) a requirement for adherens junctions (Wang 2015), or ii) inhibition of ROCK, which has been used widely in cultured cells (Overholtzer 2007) and in vivo (Li 2015).

In conclusion, none of these data support the conclusion that club cells participate in entosis. The authors could investigate this further as suggested in point g, but should also consider other possible mechanisms, which would seem equally, if not more, likely.

3) Is cell-in-cell formation required for the alarm response?

Assuming that the authors demonstrated and defined cell-in-cell formation, the limitation remains that no attempt has been made to prove this process drives the alarm response. In the absence of experiments that manipulate cell-in-cell formation, this relationship is purely correlative.

In the entosis field, ROCK inhibition has been used to disrupt the process and explore outcomes. The authors would need to do significant further work of this kind to tie any observations on cell-in-cell formation directly to the alarm response.

Reviewer #3 (Remarks to the Author):

The paper describes a cell biological and molecular approach towards identification of the zebrafish alarm substance/Schreckstoff /fear pheromone. Karl von Frisch speculated in 1938 that an injured individual in a given fish shoal secretes an "alarm substance" which elicits flight behaviour in the other individuals. The biochemistry behind this substance and the signalling cascades involved

remain largely unknown. The work presented by Chia Shu Ming et al is interesting not only because of the biochemistry involved but also because this substance seems to control social behaviour in fish.

In the present study the authors report that bacteria present in or on zebrafish skin are getting internalized via "entosis" into club cells. There the bacteria-derived substance(s) come in contact with a host-derived (mucus?) substance forming the active "alarm substance". The alarm substance is leaking from the club cells of one individual and then induces flight behaviour in other fish. For me these are early insights into a fascinating phenomenon. Some observations appear convincing and well supported by the data. Other observations, however, need more experimental support and/or better presentation to convince the referee. Here is my view:

1. The authors show quite convincingly in Figure 3 and 4 and by using a germ-free zebrafish culture, that part of the alarm substance comes from one species of commensal /resident zebrafish bacteria, *Staphylococcus* sp.
2. It is puzzling that only one species of the microbiota seems to matter. If internalization of a "pathogen" into club cells is due to the involvement of the innate immune machinery: what is so special with *Staphylococcus*; and why are other bacteria (eg *Pseudomonas*) not eliminated as well?
3. My major concern is the experimental set-up to prove that bacteria are internalized, that "entosis" is involved, and that the active substance is put together in the club cells before getting released via leakage. Much more work seems needed to substantiate these claims. Examples of major problems:
 - a) Fig 1a shows localization of bacteria in cells of the skin. However, the resolution of the images is too low to distinguish club cells from surface cells from neutrophils. The white arrow in Fig 1a points to a cell which is certainly not showing any label and therefore does not contain bacteria. Fig 1d does not convince me that we see "the cytoplasm of two club cells diffusely labelled". To make their points, the authors might consider to add a scheme of fish skin to the Figure; and to complement the in situ hybridization images with higher resolution images including TEM.
 - b) The data presented in figure 2 are also not convincing at all. Figure 2 g and h do not show a labelled cell within a club cell. In Fig 2k it is not clear which cells are labelled. Many cells appear to be labelled. Why the authors consider "diffuse label on a subset of club cells" is a mystery to me. Again, better images with higher resolution including EM/TEM cell would seem to me essential.

Reviewer #1 (Remarks to the Author):

This work is designed to examine the role of epidermal club cell cannibalism and bacteria in generating fish alarm cues. I have worked on the Schreckstoff alarm system for the majority of my career and I am very excited by this current work. I am in favour of publication. I am primarily a behavioural and evolutionary ecologist, hence my comments are restricted to that component of the work. I have considered all of the techniques used by the authors, but I am not an expert in the methods used, hence I will defer to others to address whether the techniques are appropriate.

There are a couple of things that need to be changed prior to publication. This paper makes some rather bold statements in the abstract and the introduction (paragraph 2) that says there is no known benefit to the sender of Schreckstoff. This certainly was the case for many decades. Von Frisch identified fear substances in the late 1930's and the notion of no benefit to the sender did not change until we published 2 papers in *American Naturalist* (Mathis et al. 1995 and Chivers et al. 1996). We showed that predators were attracted to Schreckstoff (Mathis et al. 1995) and that when the second predators arrived, they would often fight with the original predator, leading to escape of the prey (Chivers et al. 1996). This secondary predator interference was clear experimental evidence of a benefit to the Schreckstoff sender. Others have documented similar findings of attraction to alarm cues (e.g. Wisenden, Lonnstedt). It is incorrect to say there are no benefits to the sender. My caution that secondary predator interference may be rare in the wild lead me to consider that alarm cues evolved in another context (i.e that they were part of the innate immune system). Much of this was inspired by Iger et al. (1994; *Cell and Tissue Research*) who found photographic evidence of lymphocytes and leucocytes within club cells. The presence of two membranes around the lymphocytes and leucocytes indicates they were engulfed by the club cells, suggesting the involvement of club cells in the immune system. You cited my original work on this (Chivers et al. *Proc B*) documenting that club cells are part of the immune system, but our other work (Halbgewachs et al. 2009, *Biological Journal of the Linnean Society*) was the most significant work linking alarm cells to immunity. Colin gave fathead minnows intraperitoneal injections of cortisol, a known immunosuppressant, or injections of a control substance (corn oil). He documented that fish exposed to cortisol had suppressed immune systems (as measured by a respiratory burst assay) and that they also reduced their investment in club cells. Until this current paper is published, this was the best evidence indicating that the club cells of Ostariophysan fishes are part of the innate immune system and that the alarm function of the cells evolved secondarily. After altering the text appropriately, I would recommend publication of this work. Congrats on a great study.

Douglas P. Chivers
Distinguished Professor
University of Saskatchewan

Thank you for your helpful comments. We have made a number of corrections to the manuscript in light of your concerns. We have emphasized in our introduction the literature surrounding the role of club cells in innate immunity (Iger et al, 1994, Halbgewachs et al 2009). We have also removed the statements about evolution from the abstract and introduction, but discuss the various proposals and experiments (including that of secondary predator interference) at the end of the paper. We hope you will find

these revisions satisfactory.

Reviewer #2 (Remarks to the Author):

In this manuscript, Dr Jesuthasan and colleagues report that bacteria contribute to the zebrafish alarm response, acting synergistically with a substance from the fish. They suggest that superficial cells can internalise mucus and bacteria, and then invade club cells, in which the alarm substance (Schreckstoff) is made and stored. The authors propose that the formation of this cell-in-cell structure occurs through entosis and is required for the alarm response, concluding that 'skin cell cannibalism and processing of bacteria generate the zebrafish fear pheromone'.

The paper is clearly written and presented and the overall concept is of interest. I have been asked to provide a perspective on the data relating to cell engulfment, so will restrict my comments to lines 86-139, 255-270 and Figure 2. Although the notion of cell cannibalism could be interesting in this context, I have several major concerns, as outlined below.

We thank the reviewer for the comments, and have revised the manuscript to address these concerns.

1) Does cell-in-cell formation occur?

The authors conclude that club cells undergo cell-in-cell formation based on images in 1c, 1e and Fig 2.

While many of these images could be consistent with cell internalisation, they are certainly not definitive. To convincingly demonstrate cell-in-cell formation, it is necessary to rigorously confirm that the internalised cell is inside, rather than on or adjacent to, its host.

It would significantly strengthen these data to show:

a) Cell-in-cell formation in 3-D. A full confocal stack of a club cell hosting another cell would help confirm full internalisation. This would be best visualised using a marker that clearly defines each cell periphery in the pair (e.g. adherens junctions or plasma membrane).

We have added several figures to demonstrate that cell-in-cell formation occurs. To help visualize this, we have provided transverse views of z-stacks. Fig. 1d shows a WGA labelled cell within a club cell. Fig. 1f shows a sample labelled with phalloidin to visualize the cell peripheral. The z-stack is provided as a movie [Movie 3].

To clarify the morphology of club cells, and their position within the skin, we have added a supplemental figure (Fig. S1) and a schematic diagram (Fig. 3b). It should be noted that club cells are essentially ovoid cells with a diameter of roughly 15 - 20 μm . This is considerably larger than the axial and lateral resolution of the confocal microscope. The cytoplasm of club cells is unlabelled by Syto9. Thus, cells that are within a club cell can be distinguished from those that are adjacent, especially when there is no overlap with the Syto9 label from adjacent cells.

b) An image acquired by electron microscopy to reveal how the internalised cell is

housed inside the club cell – is there an obvious vacuole as would be expected during entosis?

We have removed the interpretation that cells enter by entosis.

A previous TEM study has shown that internalized cells can sometimes have a second membrane, and at other times not ¹. There is no description of an obvious vacuole around the internalized cell within the club cell of any species, as far as we can determine.

In conclusion, more robust imaging data are required to prove cell-in-cell formation here. This would seem achievable, particularly as the penetration of club cells by other cell types has been reported previously by electron microscopy (Refs 15-18).

The 3D confocal stacks and transverse sections shown are consistent with cell-in-cell formation, and also consistent with TEM data from other fish species^{1,2}.

Rather than performing additional EM on zebrafish, which would only contribute incrementally to our current knowledge, we have chosen to use live imaging with fluorescent probes and transgenic lines, which allows us to identify some of the cells that can transfer material to club cells, to trace the origin of club cell contents (e.g. with the pulse chase experiments), and also to show establish motility of the internalized cell. These approaches have not been conducted on the other fish species, and allow us not only to confirm their findings that there is internalization of cells, including leukocytes, into club cells, but to also extend these EM studies by showing the live transport of mucus and bacteria into these cells

c) Can you explain why have more straightforward uptake mechanisms, such as transcytosis, been excluded as important in this context?

This was indeed an oversight. Based on the presence of isolated puncta, in addition to diffuse label, inside club cells (see Fig. 1a and Movie 1), we suggest that transcytosis is an additional mechanism for transfer of material from the surface to club cells. It is likely that invasion and transcytosis act in parallel.

2) What is the mechanism of possible cell-in-cell formation?

Cell engulfment can proceed through a variety of mechanisms, including phagocytosis, emperipolesis and entosis. In their title, the authors use the term 'cell cannibalism', while in the text they infer that apparent cell-in-cell formation occurs through entosis (lines 103-114 and 270).

Entosis is a specific form of cell engulfment, with some key features including: 1) It is a form of live cell engulfment in which one viable cell is internalised into another, 2) It is a homotypic form of cell engulfment, occurring exclusively between paired epithelial cells, 3) Mechanistically, it depends upon adherens junction formation and localised actomyosin contractility (Overholtzer 2007).

Based on the data presented, it is unclear why the authors attribute apparent cell-in-cell formation to entosis, versus any other kind of engulfment? The data do not justify this

conclusion, and there is no reason why this form of engulfment would be more likely or interesting than another.

We have removed the suggestion that this is entosis as opposed to another form of engulfment. We agree that the data do not justify the conclusion.

a) Please can you clarify whether the club cell itself bears the typical features of an epithelial cell required for entosis (e.g. adherens junctions)?

Based on label with β -catenin, this does not appear to be the case. This is shown in the revised Fig 2c.

b) Can you clarify whether all internalised cells are epithelial, as would be required for entosis? The majority of apparently internalised cells look small relative to the club cell – would this be expected of a superficial cell? Have you stained for an epithelial marker to be sure?

We have used an epithelial marker, and found that the internalized cells do not appear to be epithelial (Fig. 2j). The nucleus of the internalized cells appears different from that of club cells (e.g. see Fig. 1e), and we note that this is not homotypic engulfment.

Given that published work reports penetration of club cells by leucocytes, it would seem possible that heterotypic cell-in-cell formation may be occurring as well or instead (e.g. as can be seen between epithelial and immune human cells).

Based on the distinct morphology and motility of the internalizing cell (Fig. 2g-i), it does indeed appear that there is heterotypic cell-in-cell formation.

c) Can you clarify whether the internalised cells are all live, as would be the case in entosis? Is it not possible that some of the examples shown could be phagocytosis of apoptotic cells?

Two observations suggest that at least some of the internalized cells are alive. Firstly, we do not see the internalized cells labelled by an antibody for cleaved Caspase-3. Secondly, imaging of lifeACT-eGFP fish indicates that the internalized cells can be motile. Other data, where live imaging was done using syto9 labelled fish, confirm that the internalized cells can be motile (Fig. S3, Fig. 2g-i, Movie 5).

d) The stills shown in Fig 2a-b appear to suggest quite rapid release of bacteria into a club cell as it contacts a labelled neighbour (a copy of the movie would be more helpful). This would be unexpected with respect to entosis, in which the internalised cell is scissioned into a sealed vacuole, which only later undergoes lysosomal degradation.

A movie of this sequence is not available, unfortunately, as stacks were taken at relatively sparse intervals. The rapid release appears consistent with what has been reported with neutrophils, i.e. release of cytoplasmic contents while the cell is still outside.

e) The statement in lines 105-6 that 'entosis is characterised by a higher level of f-actin in the invading cell relative to the host' is an oversimplification. What is important is spatiotemporally restricted activation of actomyosin contractility in the internalising cell,

which pushes into its host. As such, a distinctive enrichment of actin, or phosphorylated myosin, can be seen specifically in the tail of the internalising cell (Purvanov 2014, Sun 2014). The overall intensity of actin staining is not very meaningful and certainly not diagnostic of entosis.

We apologize for this error. We do not observe localized enrichment of actin at the tail of the internalizing cell.

f) The data presented on LC3B in Figures 2k-l, and discussed in lines 110-114, have been misinterpreted. Autophagy is a recycling process that occurs in response to various stresses and is characterised by the formation of autophagosomes, often visualised as LC3-positive puncta. During entosis, a related but quite distinct, signalling pathway degrades engulfed material. In this case, LC3 is recruited directly onto the entotic vacuole. This process involves transient labelling at the surface of this large endocytic compartment with LC3 (Florey 2011), and looks nothing like the image presented in Figure 2k. Moreover, during entosis, LC3 recruits in the host cell, not within the inner cell, as shown in 2l.

We thank the reviewer for pointing out this error. Fig 2k has been removed, and the discussion of the results has been edited.

g) Neither the actin nor LC3 data define this process as entosis. If the authors wish to explore entosis they should instead investigate i) a requirement for adherens junctions (Wang 2015), or ii) inhibition of ROCK, which has been used widely in cultured cells (Overholtzer 2007) and in vivo (Li 2015).

As noted above, we have removed the claim that entosis is involved and will thus not pursue this.

In conclusion, none of these data support the conclusion that club cells participate in entosis. The authors could investigate this further as suggested in point g, but should also consider other possible mechanisms, which would seem equally, if not more, likely.

We agree with this criticism. We point out in the revised manuscript that there are other mechanisms that can underlie the cell-in-cell phenomenon observed in club cells.

3) Is cell-in-cell formation required for the alarm response?

Assuming that the authors demonstrated and defined cell-in-cell formation, the limitation remains that no attempt has been made to prove this process drives the alarm response. In the absence of experiments that manipulate cell-in-cell formation, this relationship is purely correlative.

In the entosis field, ROCK inhibition has been used to disrupt the process and explore outcomes. The authors would need to do significant further work of this kind to tie any observations on cell-in-cell formation directly to the alarm response.

Cell-in-cell formation is unlikely to be required for formation of the alarm substance. This can be deduced from our previous findings that mucus alone is sufficient. The introduction has been rewritten to provide a clearer account of the previous findings.

The main aim of this project was to test whether bacteria, which is found in mucus, could provide an alarm signal in fish. The data indicates that at least one strain can elicit alarm behaviour in zebrafish. The secondary aim was to establish whether there could be a link between the two components of the alarm substance, which are mucus and bacteria, with club cells, which have long been associated with the alarm substance. The data indicate that both are transported to club cells.

We do not wish to claim that transport via cell internalization is essential for making the alarm substance, and have thus changed the title and re-written the introduction.

References

1. Iger, Y., Abraham, M. & Wendelaar Bonga, S. E. Response of club cells in the skin of the carp *Cyprinus carpio* to exogenous stressors. *Cell Tissue Res.* 277, 485–491 (1994).
2. Chapman, G. B. & Johnson, E. G. An electron microscope study of intrusions into alarm substance cells of the channel catfish. *J. Fish Biol.* 51, 503–514 (1997).

Reviewer #3 (Remarks to the Author):

The paper describes a cell biological and molecular approach towards identification of the zebrafish alarm substance/Schreckstoff /fear pheromone. Karl von Frisch speculated in 1938 that an injured individual in a given fish shoal secretes an “alarm substance” which elicits flight behaviour in the other individuals. The biochemistry behind this substance and the signalling cascades involved remain largely unknown. The work presented by Chia Shu Ming et al is interesting not only because of the biochemistry involved but also because this substance seems to control social behaviour in fish.

In the present study the authors report that bacteria present in or on zebrafish skin are getting internalized via “entosis” into club cells. There the bacteria-derived substance(s) come in contact with a host-derived (mucus?) substance forming the active “alarm substance”. The alarm substance is leaking from the club cells of one individual and then induces flight behaviour in other fish. For me these are early insights into a fascinating phenomenon. Some observations appear convincing and well supported by the data. Other observations, however, need more experimental support and/or better presentation to convince the referee. Here is my view:

We thank the reviewer and address the comments in detail below.

1. The authors show quite convincingly in Figure 3 and 4 and by using a germ-free zebrafish culture, that part of the alarm substance comes from one species of commensal /resident zebrafish bacteria, *Staphylococcus* sp.

2. It is puzzling that only one species of the microbiota seems to matter. If internalization of a “pathogen” into club cells is due to the involvement of the innate immune machinery: what is so special with *Staphylococcus*; and why are other bacteria (eg *Pseudomonas*) not eliminated as well?

The major goal of this paper is to test the hypothesis that a substance from bacteria is capable of eliciting an alarm response. The data suggest that one species is capable of producing an effective substance, consistent with this hypothesis. Some of the other species tested had a weaker effect only. We are not certain what is special about *Staphylococcus* ZWU0021, but will pursue this in future work. Other species, which were not tested here, may also provide an equally effective substance. However, an exhaustive test of all commensals is beyond the scope of this paper.

These data further suggest that there is a pathway that transports material from the surface of the fish into club cells, which does not discriminate for specific bacteria. For example, labelled *E. coli* (revised Fig. 3a-e) and zymosan that were originally in the water were also subsequently found inside club cells. Thus, there is no specificity for *Staphylococcus* with regards to transport into club cells.

3. My major concern is the experimental set-up to prove that bacteria are internalized, that “entosis” is involved, and that the active substance is put together in the club cells before getting released via leakage. Much more work seems needed to substantiate these claims. Examples of major problems:

a) Fig 1a shows localization of bacteria in cells of the skin. However, the resolution of the images is too low to distinguish club cells from surface cells from neutrophils. The white arrow in Fig 1a points to a cell which is certainly not showing any label and therefore does not contain bacteria.

We have revised the manuscript, and provide better images and more data to support the claims that are made.

With regards to the claim that bacteria are found inside club cells, the original Fig. 1a was a maximum projection, which was intended to provide an overview. However, this made it difficult to distinguish the different cell types. To make the data more accessible, and to make the cells clearer, we have now only show single optical sections as well as transverse sections through the z-stack. Additionally, a z-stack is provided (Movie 9)

Specifically, the revised Figs. 3l and m show two different planes of a club cell in the former Fig. 1a. The arrowheads in Fig. 3l show bacteria labelled by the EUB338 probe. Figs. 3l', m' and m'' are transverse sections through the club cell, which shows the presence of labelled bacteria in the club cell. There are two puncta indicated in Fig. 3l' and in Fig. 3m'. These are visible on the electronic copy, but may not be visible on a laser printed copy.

Fig 1d does not convince me that we see "the cytoplasm of two club cells diffusely labelled". To make their points, the authors might consider to add a scheme of fish skin to the Figure; and to complement the in situ hybridization images with higher resolution images including TEM.

The original Fig. 1d is now Fig. 3a. The statement that the cytoplasm of two club cells is diffusely labelled is based on the following:

1. Club cells are large ovoid cells with a diameter of approximately 20 μm . We have added a schematic diagram that shows the different cell types (Fig. S1), so that the morphology of club cells is clearer. Phalloidin label (e.g. Fig. 1f and Fig. 2d) shows the outline of these cells.
2. Syto9 (shown here in green) labels the nucleus, which is generally located in the center of the club cell. The bulk of the cytoplasm is unlabelled by Syto9, and thus appears dark. The red arrowheads in Fig. 3a show club cells with unlabelled cytoplasm. Organelles in club cells are mainly located in a perinuclear position, as indicated by Syto9 and 4-Di-2-Asp, and previously described in other species^{1,2}.
3. There is pHrodo Red signal (shown in magenta) around the nucleus in the two club cells indicated by the yellow arrowheads, but not in surrounding cells.
4. The z-stack (Movie 5) shows that the diffuse magenta label is found at several focal planes, associated with the club cell nuclei that are indicated by the yellow arrowheads.

The best interpretation we have for the presence of the pHrodo red restricted to the volume around the club cell nucleus, but excluded from surrounding cells, is that it is in the cytoplasm of the club cell.

As suggested by the reviewer, we have included a schematic diagram (Fig. 3b), to make this point more clearly.

The intent of the original Fig. 1d was to provide data suggesting that material from exogenous bacteria can be trafficked to the cytoplasm of club cells. A pulse-chase experiment using a fluorescently labelled bacteria allows this to be done. Increasing resolution with the use of TEM will not allow this fluorescent tag to be visualized. It will allow bacteria to be seen, but the resolution of Fig. 3l-m is sufficient for this conclusion to be drawn.

b) The data presented in figure 2 are also not convincing at all. Figure 2 g and h do not show a labelled cell within a club cell.

To make this clearer, we have added transverse sections, and separated the channels. This is now shown in the revised Fig. 1 (panels g-o).

The arrowhead in Fig. 1g indicates a second cell. The club cell nucleus is located in a more shallow position. This is shown in Fig. 1j. The transverse section (Fig. 1m) shows both the club cell nucleus (cyan arrowhead) and the second cell (yellow arrowhead). Two club cells without a second cell can be seen to the left (green arrowheads).

To aid the reader, we have also included a z-stack (movie 3).

We have also added additional data showing a labelled cell within a club cell (revised Fig. 1d).

In Fig 2k it is not clear which cells are labelled. Many cells appear to be labelled. Why the authors consider “diffuse label on a subset of club cells” is a mystery to me.

This panel has been removed.

Again, better images with higher resolution including EM/TEM cell would seem to me essential. It is true that TEM provides higher resolution than fluorescence microscopy. However, we disagree that TEM is essential in the context of this paper. Club cells have a diameter of approximately 20 μm . This is considerably larger than the axial resolution of a confocal microscope, which is on the order of 1 μm , given the objective NA of 1.0, wavelength of ~500 nm and the index of refraction of water (1.33). Electron microscopy would be essential if we were addressing a question that required nanometer resolution, and if the cell being investigated were flat.

To improve resolution, we have deconvolved the z-stacks. This is now used to provide the transverse sections shown in Fig. 1m, Fig. 3l, Fig. 3m and Fig. 3m’.

One advantage of the mode of imaging that have pursued here is that live imaging is possible. This allows dynamic assessment. In addition to the imaging of a cell carrying pHrodo Red *E. coli* shown in Fig. 3e, f, and tracing of mucus, we now show several time points indicating that cells inside club cells are motile (Fig. 2g-i and Fig. S2).

References

1. Whitear, M. & Mittal, A. K. Fine structure of the club cells in the skin of ostariophysan fish. *Z. Mikrosk. Anat. Forsch.* **97**, 141–157 (1983).
2. Damasceno, E. M., Monteiro, J. C., Duboc, L. F., Dolder, H. & Mancini, K. Morphology of the epidermis of the neotropical catfish *Pimelodella lateristriga* (Lichtenstein, 1823) with emphasis in club cells. *PLoS One* **7**, e50255 (2012).

REVIEWERS' COMMENTS:

Reviewer #2 (Remarks to the Author):

In their revised manuscript, Dr Jesuthasan and colleagues report that bacteria contribute to the zebrafish alarm response, acting synergistically with a substance from the fish. They also present data on cell-in-cell formation by club cells, the source of alarm substance, suggesting this may contribute to their uptake of bacteria and mucus.

The overall conclusion of the paper, that bacteria can evoke alarm behaviour, appears to be well supported and of interest to the field.

However, the mechanistic conclusions on uptake of bacteria/mucus have changed significantly between the original and revised versions, and remain somewhat unclear.

I was asked to provide a perspective on entosis. My view was that the original data did not support a role for this process and the manuscript has been revised accordingly. The reinterpreted data actually exclude entosis as a mechanism and the term has been removed from the working model (and cell cannibalism has been taken out of the title). While these changes do address the specific concerns raised, they do not provide a definitive alternative model.

Major comments:

1) The data indicating that mucus/bacteria are internalised by club cells seems quite clear and the presentation of imaging data has been improved by the revisions. I am convinced that mucus and bacteria are present in club cells.

2) However, the insights into the mechanism of uptake are still not well developed. In the original draft the authors gave significant weight to this part of their story. Cell cannibalism was included in their title and entosis in their working model, but in the revised version this mechanism has been excluded as relevant. This is quite a dramatic change in conclusions and there is limited new data to provide an alternative insight. Reference to both transcytosis and emperipolesis has been added, based on imaging observations. While some of the data would appear consistent with this, the relationships between these processes and mucus/bacteria uptake are not robustly quantified and not experimentally manipulated, leaving only a correlation. It is also unclear to me why phagocytosis has been excluded from the assortment of possible mechanisms?

The question then is whether the broad conclusions of the paper are sufficient for publication without a more definitive mechanism, which is a matter for editorial judgment. In my view:

a) If a mechanism for uptake is key to the paper, as it seemed in the original draft, the current data still require further development.

b) If the paper stands without this detail, it would be advisable to reduce reference to uptake mechanisms further – for example, the negative data on entosis seem unnecessary (lines 85-99) and distract from the main thrust of the story.

Minor points:

1) Add graphs for some of the quantification mentioned only in the text e.g. in line 68, line 118.

2) In some figures related to cell invasion (e.g. Fig 1d, 3n), the 'internalised cell' has no obvious DNA stain – this is not a convincing example of a 'cell-in-cell'.

3) Quantifying the association between cell invasion and WGA positivity would be helpful in building evidence for a mechanistic link (line 80).

Reviewer #3 (Remarks to the Author):

The authors followed all my comments and produced a very nice paper. I am convinced that the observations will set an agenda for possible future work in this area. Thomas Bosch, Kiel.

REVIEWERS' COMMENTS:

Reviewer #2 (Remarks to the Author):

In their revised manuscript, Dr Jesuthasan and colleagues report that bacteria contribute to the zebrafish alarm response, acting synergistically with a substance from the fish. They also present data on cell-in-cell formation by club cells, the source of alarm substance, suggesting this may contribute to their uptake of bacteria and mucus.

The overall conclusion of the paper, that bacteria can evoke alarm behaviour, appears to be well supported and of interest to the field.

However, the mechanistic conclusions on uptake of bacteria/mucus have changed significantly between the original and revised versions, and remain somewhat unclear.

I was asked to provide a perspective on entosis. My view was that the original data did not support a role for this process and the manuscript has been revised accordingly. The reinterpreted data actually exclude entosis as a mechanism and the term has been removed from the working model (and cell cannibalism has been taken out of the title). While these changes do address the specific concerns raised, they do not provide a definitive alternative model.

Major comments:

1) The data indicating that mucus/bacteria are internalised by club cells seems quite clear and the presentation of imaging data has been improved by the revisions. I am convinced that mucus and bacteria are present in club cells.

We thank the reviewer for his/her feedback.

2) However, the insights into the mechanism of uptake are still not well developed. In the original draft the authors gave significant weight to this part of their story. Cell cannibalism was included in their title and entosis in their working model, but in the revised version this mechanism has been excluded as relevant. This is quite a dramatic change in conclusions and there is limited new data to provide an alternative insight. Reference to both transcytosis and emperipolesis has been added, based on imaging observations. While some of the data would appear consistent with this, the relationships between these processes and mucus/bacteria uptake are not robustly quantified and not experimentally manipulated, leaving only a correlation. It is also unclear to me why phagocytosis has been excluded from the assortment of possible mechanisms?

While we cannot exclude the possibility that phagocytosis can happen, we can find no instance of this in the fish we have examined. There is, for example, no instance of enrichment of f-actin in club cells at the site of engulfment. EM studies in other species have consistently reported intrusions into club cells, consistent with invasion.

The question then is whether the broad conclusions of the paper are sufficient for publication without a more definitive mechanism, which is a matter for editorial judgment. In my view:

a) If a mechanism for uptake is key to the paper, as it seemed in the original draft, the current data still require further development.

We think that the mechanism for uptake is not central to this paper. Rather, the focus is on the idea that bacteria contribute to the alarm substance.

b) If the paper stands without this detail, it would be advisable to reduce reference to uptake mechanisms further – for example, the negative data on entosis seem unnecessary (lines 85-99) and distract from the main thrust of the story.

We have not removed this, as this would be of interest to scientists working on immunity in the skin.

Minor points:

1) Add graphs for some of the quantification mentioned only in the text e.g. in line 68, line 118.

We have added two graphs, to document that quantification mentioned in these lines. These are provided as supplementary Fig. 2 and 4.

2) In some figures related to cell invasion (e.g. Fig 1d, 3n), the 'internalised cell' has no obvious DNA stain –

this is not a convincing example of a 'cell-in-cell'.

The DNA label of the invading cell is visible in the confocal stacks. These files have been made available in Figshare.

3) Quantifying the association between cell invasion and WGA positivity would be helpful in building evidence for a mechanistic link (line 80).

The data show that invading cells contain WGA, and that material can be passed from motile cells to club cells. A diversity of cells have been reported to enter club cells. Future experiments will be required to determine the identity of all cells that can take up mucus. Their motility can then be inhibited to assess the extent of mucus transport to club cells via cell invasion.